# Effects of Drought and Salinity on Two Commercial Varieties of *Lavandula angustifolia* Mill

**DOI:** 10.3390/plants9050637

**Published:** 2020-05-16

**Authors:** Zsolt Szekely-Varga, Sara González-Orenga, Maria Cantor, Denisa Jucan, Monica Boscaiu, Oscar Vicente

**Affiliations:** 1Faculty of Horticulture, University of Agricultural Sciences and Veterinary Medicine of Cluj-Napoca, 400372 Cluj-Napoca, Romania; zsolt.szekely-varga@usamvcluj.ro (Z.S.-V.); maria.cantor@usamvcluj.ro (M.C.); 2Institute for the Conservation and Improvement of Valencian Agrodiversity (COMAV), Universitat Politècnica de València, Camino de Vera s/n, 46022 Valencia, Spain; ovicente@upvnet.upv.es; 3Mediterranean Agroforestry Institute (IAM), Universitat Politècnica de València, Camino de Vera s/n, 46022 Valencia, Spain; sagonor@etsia.upv.es (S.G.-O.); mobosnea@eaf.upv.es (M.B.)

**Keywords:** abiotic stress, growth inhibition, ion homeostasis, *Lavandula* varieties, osmotic adjustment, salt stress, water deficit

## Abstract

Global warming is not only affecting arid and semi-arid regions but also becoming a threat to agriculture in Central and Eastern European countries. The present study analyzes the responses to drought and salinity of two varieties of *Lavandula angustifolia* cultivated in Romania. Lavender seedlings were subjected to one month of salt stress (100, 200, and 300 mM NaCl) and water deficit (complete withholding of irrigation) treatments. To assess the effects of stress on the plants, several growth parameters and biochemical stress markers (photosynthetic pigments, mono and divalent ions, and different osmolytes) were determined in control and stressed plants after the treatments. Both stress conditions significantly inhibited the growth of the two varieties, but all plants survived the treatments, indicating a relative stress tolerance of the two varieties. The most relevant mechanisms of salt tolerance are based on the maintenance of foliar K^+^ levels and the accumulation of Ca^2+^ and proline as a functional osmolyte in parallel with increasing external salinities. Under water stress, significant increases of Na^+^ and K^+^ concentrations were detected in roots, indicating a possible role of these cations in osmotic adjustment, limiting root dehydration. No significant differences were found when comparing the stress tolerance and stress responses of the two selected lavender varieties.

## 1. Introduction

One of the most significant challenges for agriculture and food security in the 21st century is adapting to the new environmental conditions triggered by global warming. In different areas of the word, especially in arid and semi-arid regions, crop yields are strongly dependent on intensive irrigation, which leads to the secondary salinization of the agricultural land. The effects of climate change, including more extended, intense, and frequent drought periods, will worsen this problem [1,2,3]. Drought and soil salinity are the most harmful environmental stress factors for plants [4]. Both negatively affect crop growth, development, and productivity through the induction of osmotic and oxidative stress, with the additional component of ion toxicity in the case of salt stress [5,6,7,8,9,10,11,12]. Plants generally respond to these stressful conditions by a reduction (or even complete inhibition) of growth [8,9,11,12], an inhibition of photosynthesis [13,14,15], and an activation of a wide array of conserved tolerance mechanisms that include the accumulation of different osmolytes for osmotic adjustment [15,16,17,18], the activation of enzymatic and non-enzymatic antioxidant systems [19,20,21,22] and the expression of specific defense proteins [23,24]. Responses specific to particular stress conditions are also activated, such as the regulation of ion transport and ion homeostasis in the case of salt stress [10,25].

The *Lavandula* genus includes more than 45 species and about 400 varieties [26,27,28]. The genus is native to the Mediterranean Basin, from southern Europe to North and East Africa and the Middle East to southwest Asia and southeast India [29,30]. Many lavender species are appreciated for their cosmetic and pharmaceutical proprieties (antimicrobial, analgesic, anti-fungal, sedative, calming, relaxing, antidepressant, anti-inflammatory, healing, regenerating, tonic and antiseptic [31,32,33,34,35,36]).

*Lavandula angustifolia* Mill. is a Mediterranean perennial species that has been known and used since ancient times [37]. Its ecological optimum is found in climates with warm, sunny summers but cold winters [38], which has permitted its cultivation outside the Mediterranean region. The woody root system can grow up to 3–4 m deep, improving water absorption in arid soils [38,39].

In Romania, this species was mainly grown in small gardens as an ornamental plant. Since 2011, however, the implementation of a European project has boosted its cultivation, now occupying up to 300 hectares, approximately, as lavender is an important source of essential oils used in the cosmetics industry. The most popular varieties in this country are: ‘Blue scent’, ‘Grosso’, ‘De Moara Domneasca’, ‘Codreanca’ and ‘Sevtopolis’ [40]. Although the species originates from warmer regions, these varieties are adapted to the climates of Romania and neighboring countries, as they were acclimatized to withstand colder conditions. Therefore, global warming can be a challenge for the further commercial cultivation of these genotypes in Central and Eastern Europe.

Responses to the water deficit and salt stresses of two varieties of lavender, *L. angustifolia* var. ‘Codreanca’ (LAC) with origin in Romania, and var. ‘Sevtopolis´ (LAS) from Bulgaria were analyzed in this study. The aim of the work was to investigate growth responses, ion transport, and the accumulation of compatible solutes under stress in lavender, as only little information is available on this species. In addition, we expected to establish whether either of the two varieties was more tolerant than the other and could, therefore, adapt better to the foreseeable environmental changes caused by climate change.

## 2. Results

### 2.1. Substrate Analysis

At the beginning of the experiments, the substrate in all pots did not show any significant differences in humidity or electrical conductivity (EC), with average values of 50% and 2 mS cm^−1^, respectively. As expected, after the 30-day treatments, these two parameters showed significant differences from the initial values. However, the patterns of variation were almost identical for the two varieties, LAC and LAS (Figure 1). The progressive increase in NaCl concentration in the irrigation water led to a parallel, concentration-dependent increase in substrate EC (Figure 1a), whereas substrate moisture was somewhat higher in the salt-irrigated pots when compared to the controls, but no significant differences were observed between the different NaCl treatments (Figure 1b); this can be explained by a reduced absorption of water by the plants in the presence of salt. Substrate water content dropped to very low levels after withholding irrigation for one month—1.4% and 3.1% in LAC and LAS, respectively (Figure 1b). At these low humidity conditions, it was not possible to measure substrate EC with the WET-2 sensor (Figure 1a).

### 2.2. Plant Growth Analysis

Considering the aerial parts of the plants, water deficit and salt treatments inhibited seedling growth when compared to the non-stressed controls, with a similar degree between the two *L. angustifolia* varieties (Figure 2). Root length, however, increased significantly (2–3 cm) in the water-stressed plants of both varieties, LAC and LAS (Figure 2a). Regarding the salt stress treatments, a slight increase in root length was detected in LAC plants—not in the LAS variety—but it was statistically significant only at the highest salinity tested, 300 mM NaCl (Figure 2a).

Elongation of the stems during the 30-day treatments was significantly larger in the control than in the stressed plants, with small differences observed between the water deficit and the three salt treatments. These differences were non-significant in LAC plants, whereas the relative reduction in stem elongation (SE) was more pronounced at high salinities (200 and 300 mM NaCl) in the LAS variety (Figure 2b). The effect of the stress treatments was also evident, considering the relative increase in stem diameter (SD), which showed a significant, concentration-dependent reduction in salt-treated plants with respect to the controls (Figure 2c). This effect was much more prominent in the plants subjected to water stress, for which negative values were calculated for this parameter; that is, stem diameter actually decreased during the treatment in both varieties, especially in LAS plants (Figure 2c). Regarding the increase in the number of branches, a relative reduction with respect to the corresponding controls was observed in parallel to the rise in external salinity, as well as in water-stressed plants, for both varieties (Figure 2d).

Once the stress treatments were concluded, the collected plant samples were used to determine the most relevant parameters to assess the degree of growth inhibition in response to water deficit and salt stress, namely, the fresh weight and water content of roots, stems, and leaves (Figure 3). A sharp reduction in root fresh weight (FW) was observed in plants exposed to water stress (Figure 3a), which was associated with dehydration of the roots (Figure 3b). The NaCl treatments, on the other hand, did cause inhibition of root growth in a concentration-dependent manner (Figure 3a), but not any significant water loss (Figure 3b). As in roots, water deficit caused a substantial reduction of FW in stems and leaves (Figure 3c,e), accompanied by a significant decrease in water content (Figure 3d,f). Salt stress affected the growth of the aerial parts of the plants more than that of the roots. The FW of stems and leaves decreased in parallel with increasing external NaCl concentrations, reaching similar values than in the water stress treatment in the presence of 300 mM NaCl (Figure 3c,e). Contrary to roots, salt stress caused a progressive reduction of WC in stems and leaves (Figure 3d,f), although, in general, it was not as pronounced as in the water-stressed plants.

When comparing the two lavender varieties, no significant differences were detected in any of the measured parameters between LAC and LAS plants that underwent the water deficit and salt stress treatments.

### 2.3. Photosynthetic Pigments

Leaf contents of photosynthetic pigments—chlorophylls a (Chl a, Figure 4a) and b (Chl b, Figure 4b), and carotenoids (Caro, Figure 4c)—were determined in plants of the two selected lavender varieties, LAC and LAS, after the applied treatments. Chl a and Chl b contents decreased significantly (although Chl b only slightly) under water stress conditions in both varieties (Figure 4a,b). Chl a average values were also reduced in response to increasing salinity, but the differences with the control were not statistically significant in most cases—except at high salinities in the LAS variety – due to the variability in the assayed individual replicas causing large SE values. Here again, both varieties responded in a similar way to the salt treatments (Figure 4a). A reduction of mean Chl b levels with respect to the control was also observed in the presence of 100 and 200 mM NaCl, to increase over the control values at the highest salt concentration tested, especially the LAC variety (Figure 4b). Total carotenoid contents were very low and did not differ much between the different treatments except for a relatively large reduction observed at high salinities (Figure 4c).

### 2.4. Ions Content Variation in Response to Stress

The concentrations of Na^+^, Cl^−^, K^+^, and Ca^2+^ were measured in roots, stems, and leaves of plants subjected to the water deficit and salt stress treatments, together with the corresponding non-stressed controls. In all cases, the patterns of variation of the different ions were practically identical for the two investigated *L. angustifolia* varieties, LAC and LAS (Figure 5 and Figure 6).

As expected, Na^+^ concentration increased in response to the salt treatments in the three organs analyzed. In roots, Na^+^ reached roughly the same concentration at all external salinities tested, approximately 1 mmol g^−1^ DW (dry weight) (Figure 5a), and an equal value was determined in stems (Figure 5b) and leaves (Figure 5c) in the presence of 100 mM NaCl. In plants treated with 200 or 300 mM NaCl, Na^+^ accumulated to higher levels in stems (Figure 5b) and even (slightly) higher in the leaves, reaching ~2 mmol g^−1^ DW as the maximum value (Figure 5c). Interestingly, water stress induced a significant increase of Na^+^ content, two- to three-fold higher than in the controls, specifically in the roots (Figure 5a) but not in the aerial part of the plants.

Stress-induced changes in Cl^−^ contents followed a pattern similar to that of Na^+^, with some quantitative differences. Cl^−^ increased in response to increasing external salinity in a concentration-dependent manner also in roots (Figure 5d), not only in the aerial parts of the plants. Also, under all tested conditions Cl^−^ concentrations were lower than those of Na^+^, reaching maximum values of, approximately, 350, 600, and 800 µmol g^−1^ DW in roots, stems, and leaves, respectively (Figure 5d–f). Moreover, no increase over Cl^−^ control levels was detected as a response to water stress in any plant organ.

Mean K^+^ contents in roots of non-stressed plants (~300 µmol g^−1^ DW) increased significantly, almost twofold, in response to water deficit, whereas a slight but significant decrease was observed in salt-treated plants (Figure 6a). Control K^+^ concentrations in stems and leaves were about double than those in roots and did not change significantly (or increased only slightly) in stems and leaves, respectively, of water-stressed plants (Figure 6b,c). Changes in K^+^ concentrations in response to salt stress, however, differed in stems and leaves. A substantial reduction was observed in stems of plants treated with 100 and 200 mM NaCl, but K^+^ concentration increased again, reaching control levels at the highest salt concentration tested, 300 mM NaCl (Figure 6b). However, no salt-dependent K^+^ reduction was detected in leaves; on the contrary, a small increase over the control value was observed in the presence of salt (Figure 6c).

Regarding Ca^2+^ concentrations, similar qualitative patterns of variation in response to the water and salt stress treatments were observed in roots, stems, and leaves. In non-stressed control plants, Ca^2+^ contents were similar in roots and stems, ~20 µmol g^−1^ DW, but about three-fold higher in leaves (Figure 6d–f). Water stress did not induce any significant change in Ca^2+^ concentration in roots or stems (Figure 6d,e), and only caused a small increase in leaves (Figure 6f). Salt stress, on the other hand, caused a significant increase in Ca^2+^ levels in the three organs, which was roughly in parallel with the increasing NaCl concentration in the irrigation water. Absolute Ca^2+^ concentrations under the same salinity were higher in stems and, especially, in leaves than in roots, reaching maximum values of about 250 µmol g^−1^ DW (Figure 6f). As for the rest of the measured ions, no significant differences were observed in Ca^2+^ levels between plants of the two selected varieties, LAC and LAS, when subjected to the same treatment; however, the calculated mean values were generally lower for the Bulgarian LAS variety (Figure 6d–f).

### 2.5. Osmolyte Levels

In this study, the accumulation of proline (Pro) and total soluble sugars (TSSs)—two common plant organic osmolytes—was determined in leaves of lavender plants exposed to water deficit and salt stress treatments (Figure 7). Pro contents did not increase in response to water stress in either of the two selected *L. angustifolia* varieties, and even decreased slightly (but significantly) in LAS (Figure 7a). The salt treatments, on the contrary, induced a significant increase in Pro concentration in both varieties, although reaching higher average values (up to ~120 µmol g^−1^ DW) in LAC than in LAS plants (Figure 7a). However, due to the relatively large SE of the means, the differences between the two varieties were not statistically significant.

Significant differences between both varieties were more clearly detected when comparing the stress-induced accumulation of TSSs (Figure 7b). In LAC plants, leaf TSS contents increased significantly in response to both water deficit and salt stress, albeit in the latter case only in the presence of 100 or 200 mM NaCl, not at the highest salinity tested (300 mM). In LAS plants, on the contrary, no changes with respect to the non-stressed control were observed for any of the applied stress treatments (Figure 7b).

### 2.6. Multivariate Analysis

A two way-ANOVA was performed to establish the effects of the treatment, varieties, and their interactions on all analyzed parameters (Table 1). The results of this analysis revealed that the ‘treatment’ factor had a highly significant effect on all measured variables, except for K^+^ and TSS levels in leaves, which did not vary considerably in response to the applied treatments. No significant effect was observed for ‘variety’, supporting the general conclusion of the individual experiments that both varieties responded to stress in the same way; the only exception was found for root length (RL), which showed a stronger increase under water deficit stress in one of the varieties, LAS. The interaction of the two factors, treatment and variety, was significant only for a few variables, namely, stem water content, root length, Na^+^ in leaves, and Cl^−^ in roots (Table 1).

A principal component analysis (PCA) was also performed, including all parameters analyzed in the plants. Of the 27 components detected, 13 had an eigenvalue higher than 1. The biplot of the two main principal components, explaining 60.4% of the total variability, is shown in Figure 8. The first component (*X*-axis), explaining 36.5% of the variability, was mostly related to the responses to salt stress, especially the accumulation of ions, whereas the second component (*Y*-axis), explaining an additional 23.9% of variability, separated responses in control plants from those of the plants subjected to water stress. For example, variables that increased significantly under salt stress, such as Na^+^, Cl^−^, Ca^2+^, and Pro, are grouped on the right part of the graph (Figure 8a). On the other hand, those that decreased strongly under water deficit conditions, including several growth parameters (e.g., FW and WC of roots, stems, and leaves) or Chl a, are located on the upper part of the graph; these were opposed to root length and K^+^ contents, which increased under salt stress (Figure 8a).

The 50 individual plants analyzed in the present work were dispersed onto the two axes of the PCA scatterplot, according to the treatments applied. A clear separation of the control, water stress, and different salt treatments was observed, with some overlapping only between the samples treated with 200 and 300 mM NaCl (Figure 8b). On the contrary, there was no separation of the two varieties; for each treatment, LAC and LAS individuals appeared mixed (Figure 8b), confirming the similar behaviors of the two varieties in their responses to stress.

## 3. Discussion

### 3.1. Plant Growth and Photosynthetic Pigments

Growth inhibition is a typical and fast response of plants exposed to environmental stress factors, such as drought and salinity, both causing osmotic stress in plants and, as a first consequence, reducing cell turgor and expansion [25]. However, growth inhibition under stress is ultimately associated with the reallocation of plant resources, which normally are used for primary metabolism and growth (i.e., biomass accumulation) towards the activation of defense mechanisms [25,41,42]. Several growth parameters were determined in water- and salt-stressed *L. angustifolia* plants, including the reduction of fresh weight and water content in roots, stems, and leaves—which is probably the most reliable way to assess growth inhibition. The data obtained clearly showed that one month of water deficit or salt treatments significantly reduced growth in the two selected lavender varieties. The decrease in FW was more pronounced in response to water stress than to high salinity, but this effect was largely due to the stronger dehydration observed in the first case. It should be pointed out, that all plants survived the harsh conditions of our experiments, 30 days of total irrigation withholding, and watering with NaCl solutions up to 300 mM. These data indicate that both *L. angustifolia* varieties are relatively resistant to water and salt stress, at least much more tolerant than most cultivars of conventional crops like *Phaseolus* [43,44]. Previous reports have also shown a reduction of growth in *L. angustifolia* and other species of the genus under both water deficit [45,46,47] and salt stress conditions [47,48,49,50,51], although in those experiments shorter treatment times and lower salt concentrations were generally used. Interestingly, root length increased significantly in water-stressed plants of both varieties. This behavior probably mimics that of the plants in nature, where roots are stimulated to grow under low moisture conditions to reach deeper and more humid soil layers.

There is strong evidence that photosynthetic pigments (chlorophylls and carotenoids) are profoundly affected by salt and water stress in many cultivated species [20,52,53,54,55], but not (or less so) in naturally tolerant wild species, such as some halophytes [56]. Therefore, the reduction of photosynthetic pigment levels, especially of chlorophylls, is generally regarded as a useful stress biomarker in plants. Previously published reports on different species of aromatic plants [57], including *L. angustifolia* [51,58], exposed to water and salt stress treatments, have also demonstrated a decrease in photosynthetic pigments. These data have been confirmed in our experimental conditions, as we show here a significant reduction of chlorophyll a and b contents in response to water deficit and salt stress. Carotenoid concentrations also decreased significantly at high external salinities, but their absolute values were very low; in practical terms, these compounds do not seem to be appropriate stress indicators in lavender.

### 3.2. Ion Accumulation

A general strategy of glycophytes (which include all major crops) to cope with high soil salinity is based on limiting the accumulation of toxic ions—fundamentally Na^+^ and Cl^−^—in the leaves, either by reducing their uptake at the root level or by blocking their transport to the aerial parts of the plant. This mechanism, however, is effective only at relatively low salt concentrations, depending on the tolerance of the particular genotype [59,60]. In the two selected lavender varieties, the salinity threshold to avoid the accumulation of high Na^+^ concentrations in the leaves appeared to be around 100 mM NaCl. Under these conditions, Na^+^ contents were the same in roots, stems, and leaves, but at higher salinities its levels in leaves increased to double those measured in the roots. Mechanisms reducing Cl^−^ uptake by the roots seemed to be efficient enough to maintain its concentrations below those of Na^+^ at all external salinities and in the three organs. However, Cl^−^ transport to stems and leaves was not blocked under any NaCl concentration, remaining always higher in stems and (especially) leaves than in roots.

Na^+^ accumulation is generally accompanied by a reduction of K^+^ levels, as both ions compete for the same transport systems [61,62]. K^+^ plays an essential role in plant metabolism, being involved in enzyme activation, osmotic adjustment and turgor generation, regulation of membrane potential, and cytoplasmic pH homeostasis [63]. A significant reduction of K^+^ concentration was detected in the roots of salt-treated lavender plants, as well as in stems in the presence of 100 and 200 mM NaCl. However, in stems of plants treated with 300 mM NaCl and in leaves at all external salinities, K^+^ contents were similar to the controls, indicating the activation of K^+^ transport to the leaves, increasing the K^+^/Na^+^ ratio and partly compensating the adverse effects of high Na^+^ concentrations. The same mechanism has been described for other species [64,65,66], and most likely contributes to the (relative) salt tolerance of the lavender varieties. Supporting this notion, recent publications can be cited highlighting the importance of K^+^ homeostasis and the maintenance of a relatively high K^+^/Na^+^ ratio in leaves of *L. stoechas* [67], or the fact that foliar K^+^ application ameliorates the responses to salt stress in *L. angustifolia* [51].

Another mechanism contributing to salt tolerance in *L. angustifolia* is the accumulation of Ca^2+^ ions in response to increasing external salinity, in a concentration-dependent manner and in all three organs, although reaching the highest concentrations in the leaves. This conclusion is supported by the well-established role of Ca^2+^ counteracting the deleterious effects of salt stress in plants [68,69].

No significant changes in ion concentrations in the plants should have been expected in response to the water deficit treatment, and this is indeed what we generally observed in our experiments, with two remarkable exceptions: the substantial accumulation (two- to three-fold over the well-watered controls) of Na^+^ and K^+^ in roots (but not in stems or leaves) of the two lavender varieties. This result indicates the presence of mechanisms for the active uptake of these cations from a low-salinity substrate, probably to contribute to osmotic adjustment in the roots and limit the strong dehydration caused by water stress.

### 3.3. Osmolyte Synthesis

The essential role of osmolytes is to limit water loss and maintain turgor in leaf cells under different stress conditions with an osmotic component, such as drought and salinity. Osmolytes, however, have additional functions as low-molecular-weight chaperons, stabilizing cell membranes and enzymes or playing a protective role against oxidative stress as ‘reactive oxygen species’ (ROS) scavengers [70,71]. One of the most common osmolytes in plants is Pro, and often a high accumulation of Pro in leaves contributes directly to salt tolerance [72]. Previous studies have reported the increase of Pro concentrations in response to salt stress in many different species, including some aromatic plants such as sage [73] or spearmint [74]. The role of Pro in the genus *Lavandula* is not yet clear. For example, a salt-induced decrease in Pro levels has been reported for *L. multifida* [75], whereas it has been considered as an indicator of water stress in *L. pedunculata* [45]. In *L. angustifolia*, an increase of Pro contents (as compared to intact plants) has been reported under salt stress [51] and in in vitro regenerated microplants [76]. Our results clearly show a significant increase in leaf Pro levels in response to salt stress. Moreover, the concentrations reached are high enough to have a relevant osmotic effect in the defense reaction against high salinity. However, Pro does not seem to play any functional role in water-stressed plants.

Soluble sugars are also common osmolytes in many plant species, playing critical functional roles in abiotic stress responses [77]. Total soluble sugar accumulation in response to salt treatments has been reported in sage [41], fennel [78], and *Satureja hortensis* L. [79]. In the present experiments, TSS levels varied significantly under water deficit and salt stress conditions, but only in one of the tested *L. angustifolia* varieties, LAC, whereas in LAS no differences could be observed with respect to control. Other reports on this species also did not find significant differences in TSS contents between control and water-stressed plants [46].

### 3.4. Comparison between the Two Lavender Varieties

The results of the present study clearly indicate that the responses to water deficit and salt stress of the two selected *L. angustifolia* varieties, LAC and LAS, were very similar, if not identical. The multivariate analysis of the data strongly supports this idea. The two-way ANOVA indicated that the ‘variety’ factor did not have a significant effect on any relevant measured variable, whereas the ‘treatment’ effect was highly significant for practically all determined parameters. Similarly, in the PCA scatterplot, the different treatments, but not the two varieties, were clearly separated. However, looking at the changes in the mean values of some relevant growth parameters, particularly the FW of roots, stems, and leaves as well as the average chlorophyll a contents, it would seem that *L. angustifolia* var. ‘Codreanca’ (LAC) was slightly more tolerant to moderate and high salt concentrations than *L. angustifolia* var. ‘Sevtopolis’ (LAS). This could be explained by the somewhat higher levels of Ca^2+^ ions and Pro accumulated in LAC plants in response to the salt stress treatments. In any case, in practical terms, the differences are too small to allow proposing any of the two varieties as more suitable to be grown under non-optimal conditions (e.g., in marginal soils or using limited or low-quality salinized water for irrigation).

## 4. Materials and Methods

### 4.1. Plant Material, Seedling Growth and Stress Treatments

Two commercial lavender varieties were used in the present work, *L. angustifolia* Mill. var. ‘Codreanca’ (LAC) and *L. angustifolia* var. ‘Sevtopolis’ (LAS). *L. angustifolia* var. ‘Codreanca’, approved in 1992, is considered as one of the best varieties for the climatic conditions of Romania, as it is resistant to frost. It produces silver-blue leaves and blue-purple, sometimes deep blue flowers that flower in May–June, reaching a height of up to 60 cm. *L. angustifolia* var. ‘Sevtopolis’ is native to Bulgaria and has a great commercial value due to its chemical composition. The plants, which grow to 40–60 cm high, are suitable for sunny or partly shaded environments. The flowers are blue and the flowering period is May–July.

Seeds of both varieties were sown in a mixture of peat (50%), perlite (25%), and vermiculite (25%). Seedlings were grown individually in 0.3 L pots, placed in plastic trays separately for each treatment, and maintained in a controlled climate chamber at the Institute for the Conservation and Improvement of Valencian Agrodiversity (COMAV), Universitat Politècnica de València, under artificial light, with a long-day photoperiod (16 h light and 8 h dark), at a constant temperature of 23 °C and 60–80% relative humidity. The pots were watered every third day with Hoagland’s nutrient solution [80] until the treatments were started, 10 weeks after sowing. Salt treatments were applied by watering the plants twice a week with an aqueous salt solution with final concentrations of 0 (control), 100, 200, and 300 mM NaCl, adding 1 L per tray. For water deficit stress (WS) treatments, irrigation was ceased entirely. Five individual plants of each variety were used per treatment as biological replicas. After the 30 days treatment, plants were harvested, and leaves, stems, and roots were collected separately. Fresh plant material was frozen in liquid N_2_ and stored at −70 °C, whereas dry material was kept at room temperature in tightly closed containers. Humidity (% vol) and electric conductivity (EC) of the pot substrates were measured at the beginning, during (not shown) and at the end of the treatments, with a WET-2 Sensor (Delta-T Devices, Cambridge, England) [56].

### 4.2. Plant Growth Parameters

At day 0, when the treatments started, the stem length and diameter, and the number of branches were determined for all plants; at the end of the treatments, these parameters were measured again. After harvesting, the root lengths (RL) and the fresh weights (FWs) of leaves, stems, and roots were measured in all plants. To obtain the water content percentages, part of the fresh material of leaves, stems, and roots was weighed (FW), dried in an oven at 60 °C for 72 h, then weighed again (DW). The water content (WC%) was calculated by the following formula:WC% = [(FW − DW)/FW] × 100

### 4.3. Photosynthetic Pigments

Chlorophyll a (Chl a), chlorophyll b (Chl b), and total carotenoids (Caro) were measured as previously described [81]. Fresh leaf material (50–100 mg) was ground in a mortar in the presence of liquid N_2_, and extracted with 1-mL ice-cold 80% (*v*/*v*) acetone, by shaking overnight in the dark at 4 °C. The samples were centrifuged at 12,000 rpm, at 4 °C, and the absorbance of the supernatant was determined at 470, 645, and 663 nm. Chlorophylls and carotenoid concentrations were calculated using the following equations [81]:Chl a (µg mL^−1^) = 12.21 (A_663_) − 2.81 (A_646_);
Chl b (µg mL^−1^) = 20.13 (A_646_) − 5.03 (A_663_);
Caro (µg mL^−1^) = (1000 A_470_ − 3.27 [chl a] − 104 [chl b])/227.

Concentrations of the photosynthetic pigments were finally expressed in mg g^−1^ DW.

### 4.4. Ion Accumulation

Sodium (Na^+^), potassium (K^+^), calcium (Ca^2+^), and chloride (Cl^−^) ions were measured in leaves, stems, and roots after the stress treatments were concluded. Extracts were prepared as described [82] from 50–100 mg of leaf material, ground to a fine powder, and suspended in 15 mL of Milli-Q water. The samples were heated for 1 h in boiling water, cooled on ice, and filtered through a nylon filter of 0.45 µm. Cation (Na^+^, Ca^2+^, and K^+^) concentrations were determined with a PFP7 flame photometer (Jenway Inc., Staffordshire, UK), and Cl^−^ was quantified with a chloride analyzer (Sherwood, model 926, Cambridge, UK).

### 4.5. Osmolyte Quantification

Proline and total soluble sugars, two main types of plant organic osmolytes, were analyzed in fresh leaf material. Proline (Pro) concentration was quantified according to the ninhydrin-acetic acid method [83]. Briefly, extracts were prepared by grinding the leaf material (0.15 g) in 2 mL of a 3% (*w*/*v*) sulfosalicylic acid solution; the samples were mixed with acid ninhydrin, incubated 1 h at 95 °C in a water bath, cooled to room temperature, and extracted with toluene. The absorbance of the organic phase was measured at 520 nm, using toluene as the blank. Reaction mixtures containing known Pro concentrations were run in parallel to obtain a standard curve. Leaf Pro contents were finally expressed in μmol g^–1^ DW.

Total soluble sugars (TSSs) were determined according to a previously described procedure [84], grinding about 0.1 g of fresh leaf material in liquid N_2_, followed by extraction with 3 mL 80% (*v*/*v*) methanol and centrifugation at 12,000 rpm for 10 min; supernatants were collected and diluted 10-fold with water. Concentrated sulphuric acid and 5% phenol were added to each sample, and the absorbance was measured at 490 nm. The TSS concentrations were expressed as equivalents of glucose, used as the standard (mg eq. glucose g^−1^ DW).

### 4.6. Statistical Analysis

Data were analyzed using Statgraphics Centurion XVI (Statgraphics Technologies, The Plains, VA, USA). Before the analysis of variance, a Shapiro–Wilk test was used to check for the validity of normality assumption and a Levene’s test was used for the homogeneity of variance. If ANOVA requirements were met, significant differences among treatments were tested by one-way ANOVA at the 95% confidence level, and post hoc comparisons were made using a Tukey HSD test. All mean values throughout the text, followed by their SE, are based on five biological replicas per variety and per treatment. All the parameters measured in plants of the control and stress treatments were subjected to multivariate analysis through a principal component analysis (PCA).

## 5. Conclusions

The present study provides new experimental data on the responses of *L. angustifolia* to drought and salinity, using two varieties, ‘Codreanca’ and ‘Sevtopolis’, common in Romania, for which very little information is available. Although plant growth was inhibited, all plants survived the harsh stress conditions applied: one month of complete withholding of irrigation and watering with NaCl at concentrations up to 300 mM, indicating that the two varieties are relatively tolerant to both types of stress, or at least more tolerant than most common crops. Salt tolerance mostly depended on the maintenance of K^+^ levels in leaves of salt-treated plants, limiting the reduction of the K^+^/Na^+^ ratio, on the salt-dependent accumulation of Ca^2+^, which partly counteracts the deleterious effects of salt stress, and on the biosynthesis and accumulation of Pro as a functional osmolyte. Interestingly, water stress induced a significant increase of Na^+^ and K^+^ concentrations in roots, which can contribute to osmotic adjustment and limit root dehydration under water deficit conditions. Although the two varieties selected for this work did not differ significantly in terms of stress tolerance, the work presented here opens the possibility to use biochemical stress biomarkers, such as Pro, for the rapid screening and eventual selection of lavender genotypes more resistant to salinity and, therefore, better adapted to a climate change scenario.

## Figures and Tables

**Figure 1 plants-09-00637-f001:**
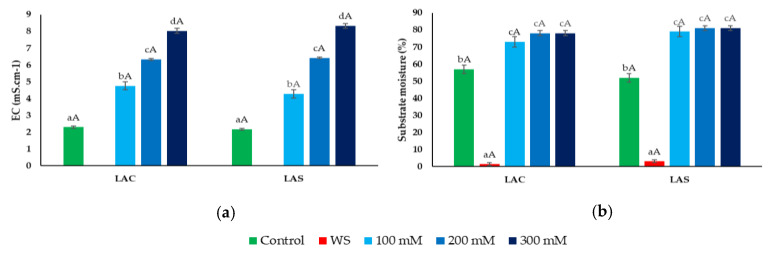
Substrate electrical conductivity (EC) (**a**), and moisture (%) (**b**), after 30 days of control, water deficit (WS) or salt stress (100, 200, 300 mM NaCl) treatments. Values shown are means ± SE (*n* = 5). Different lowercase letters above the bars indicate significant differences between treatments for each variety (LAC: *Lavandula angustifolia* var. Codreanca; LAS: *L. angustifolia* var. Sevtopolis), and different uppercase letters indicate significant differences between the two varieties for pots that underwent the same treatment, according to the Tukey test (α = 0.05).

**Figure 2 plants-09-00637-f002:**
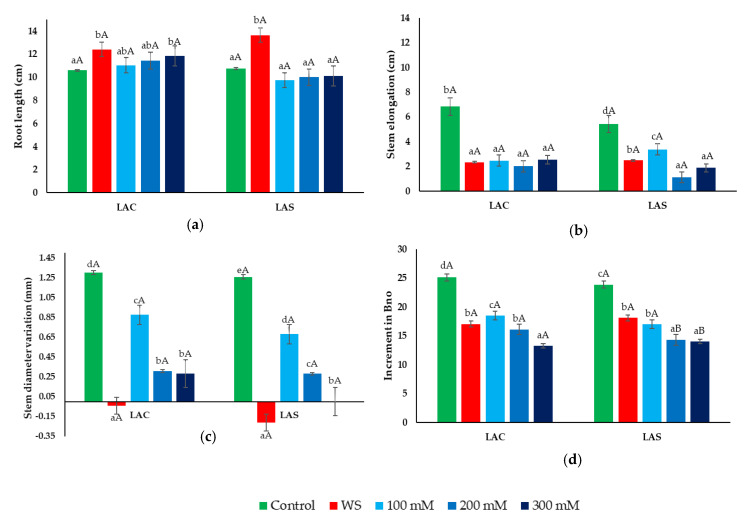
Effect of water and salt stress treatments on growth parameters in *L. angustifolia* var. ‘Codreanca’ (LAC) and var. ‘Sevtopolis’ (LAS). Root length (RL) (**a**); stem elongation (SE) (**b**); increase in stem diameter (SD) (**c**); increment in the number of branches (Bno) (**d**) in lavender plants after 30 days of growth under control conditions, in the presence of the indicated NaCl concentrations or subjected to water stress (WS) (completely withholding irrigation). Stem length, stem diameter, and number of branches were measured in all plants just before starting the treatments (time 0), and before collecting the samples (time 30). Bars represent means ± SE (*n* = 5). Different lowercase letters above the bars indicate significant differences between treatments for each variety, and different uppercase letters indicate significant differences between the two varieties for plants undergoing the same treatment, according to the Tukey test (α = 0.05).

**Figure 3 plants-09-00637-f003:**
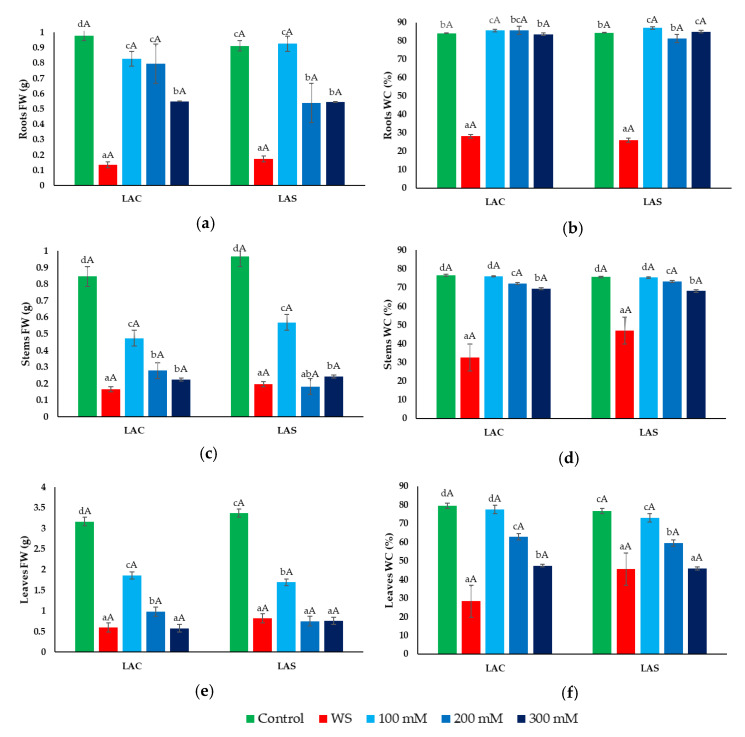
Effect of water and salt stress treatments on growth parameters in *L. angustifolia* var. ‘Codreanca’ (LAC) and var. ‘Sevtopolis’ (LAS). Fresh weight (FW) (**a**,**c**,**e**) and water content percentage (WC%) (**b**,**d**,**f**) of roots (**a**,**b**), stems (**c**,**d**) and leaves (**e**,**f**) of lavender plants after 30 days of growth under control conditions, in the presence of the indicated NaCl concentrations or subjected to water stress (WS) (completely withholding irrigation). Bars represent means ± SE (*n* = 5). Different lowercase letters above the bars indicate significant differences between treatments for each variety, and different uppercase letters indicate significant differences between the two varieties for plants undergoing the same treatment, according to the Tukey test (α = 0.05).

**Figure 4 plants-09-00637-f004:**
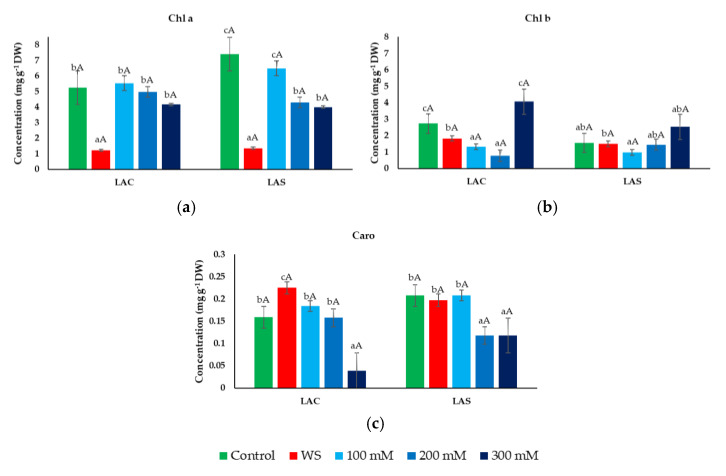
Photosynthetic pigment concentrations under water deficit and salt stress conditions. Leaf contents of chlorophyll a (Chl a) (**a**), chlorophyll b (Chl b) (**b**), and total carotenoids (Caro) (**c**) in *L. angustifolia* var. ‘Codreanca’ (LAC) and var. ‘Sevtopolis’ (LAS) plants after 30 days of growth under control conditions, in the presence of the indicated NaCl concentrations or subjected to water stress (WS) (completely withholding irrigation). DW, dry weight. Bars represent means ± SE (*n* = 5). Different lowercase letters above the bars indicate significant differences between treatments for each variety, and different uppercase letters indicate significant differences between the two varieties for plants undergoing the same treatment, according to the Tukey test (α = 0.05).

**Figure 5 plants-09-00637-f005:**
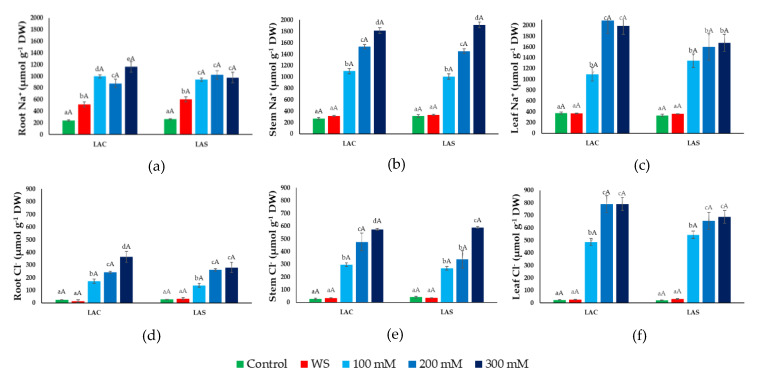
Ion contents under water deficit and salt stress conditions. Sodium (Na^+^) (**a**,**b**,**c**) and chloride (Cl^−^) (**d**,**e**,**f**) concentrations in roots (**a**,**d**), stems (**b**,**e**), and leaves (**c**,**f**) of *L. angustifolia* var. ‘Codreanca’ (LAC) and var. ‘Sevtopolis’ (LAC) plants after 30 days of growth under control conditions, in the presence of the indicated NaCl concentrations or subjected to water stress (WS) (completely withholding irrigation). DW, dry weight. Bars represent means ± SE (*n* = 5). Different lowercase letters above the bars indicate significant differences between treatments for each variety, and different uppercase letters indicate significant differences between the two varieties for plants undergoing the same treatment, according to the Tukey test (α = 0.05).

**Figure 6 plants-09-00637-f006:**
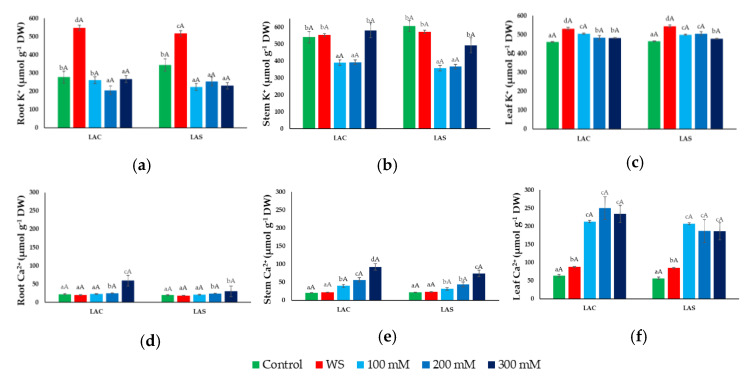
Ion contents under water deficit and salt stress conditions. Potassium (K^+^) (**a**,**b**,**c**) and calcium (Ca^2+^) (**d**,**e**,**f**) concentrations in roots (**a**,**d**), stems (**b**,**e**), and leaves (**c**,**f**) of *L. angustifolia* var. ‘Codreanca’ (LAC) and var. ‘Sevtopolis’ (LAS) plants after 30 days of growth under control conditions, in the presence of the indicated NaCl concentrations or subjected to water stress (WS) (completely withholding irrigation). DW, dry weight. Bars represent means ± SE (*n* = 5). Different lowercase letters above the bars indicate significant differences between treatments for each variety, and different uppercase letters indicate significant differences between the two varieties for plants undergoing the same treatment, according to the Tukey test (α = 0.05).

**Figure 7 plants-09-00637-f007:**
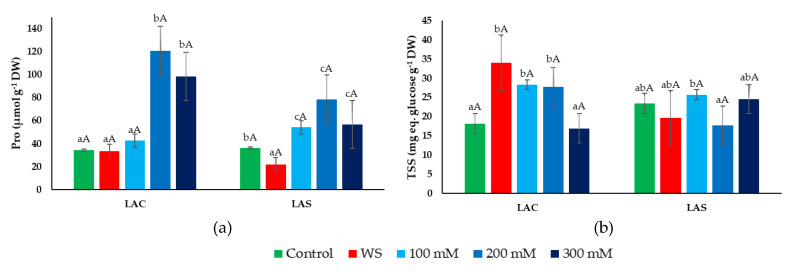
Leaf contents of proline (Pro) (**a**) and total soluble sugars (TSSs) (**b**) in *L. angustifolia* var. ‘Codreanca’ (LAC) and var. ‘Sevtopolis’ (LAS) plants after 30 days of growth under control conditions, in the presence of the indicated NaCl concentrations or subjected to water stress (WS) (completely withholding irrigation). DW, dry weight. Bars represent means ± SE (*n* = 5). Different lowercase letters above the bars indicate significant differences between treatments for each variety, and different uppercase letters indicate significant differences between the two varieties for plants undergoing the same treatment, according to the Tukey test (α = 0.05).

**Figure 8 plants-09-00637-f008:**
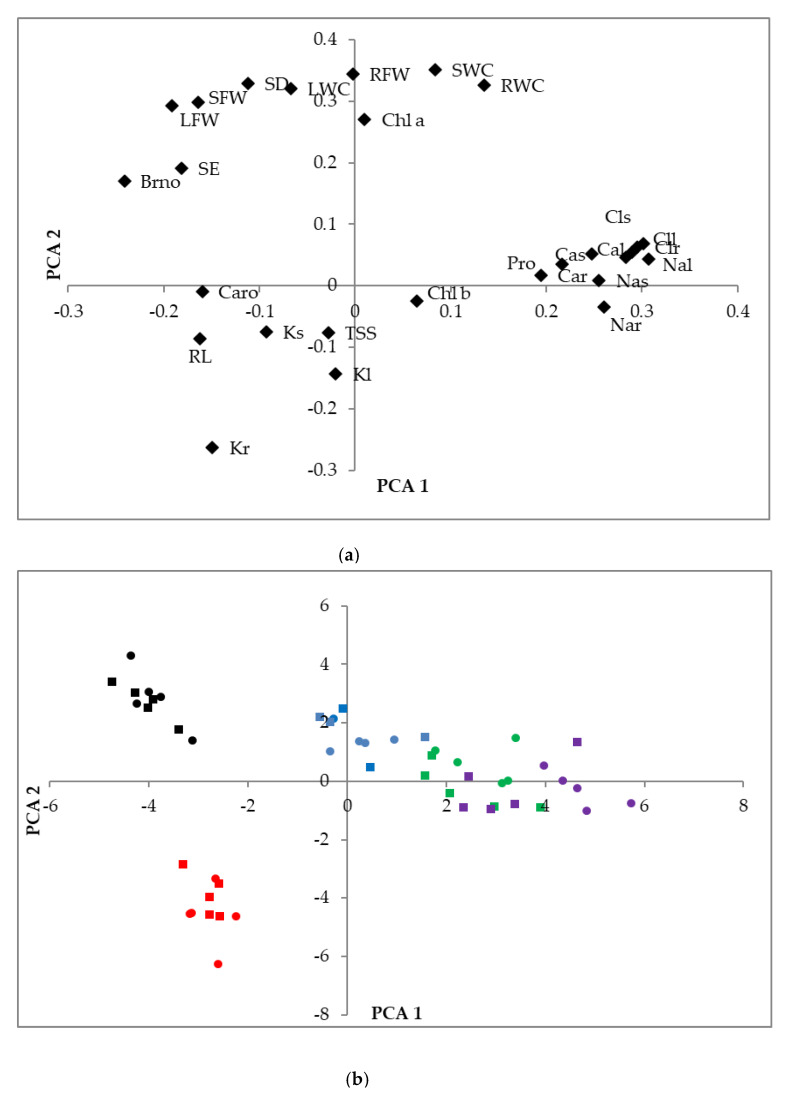
Loading plots of the Principal Component Analysis (PCA) (**a**), conducted with the analyzed traits in *L. angustifolia* (var. ‘Codreanca’ and var. ‘Sevtopolis’), subjected to salt and water stress conditions for 30 days, based on the first and second components accounting for 36.5% and 23.9%, respectively, of the total variability observed. RL, root length; SE, stem elongation; SD, stem diameter increment; Bno, increment in the number of branches; RFW, root fresh weight; RWC, root water content; SFW, stem fresh weight; SWC, stem water content; LFW, leaf fresh weight; LWC, leaf water content; Chl a, chlorophyll a; Chl b, Chlorophyll b; Caro, total carotenoids; Nar, sodium in roots; Clr, chloride in roots; Kr, potassium in roots; Car, calcium in roots; Nas, sodium in stems; Cls, chloride in stems; Ks, potassium in stems; Cas, calcium in stems; Nal, sodium in leaves; Cll, chloride in leaves; Kl, potassium in leaves; Cal, calcium in leaves; Pro, proline; TSSs, total soluble sugars. Scatter plot of the PCA scores (**b**): Control (black symbols), water stress (red), 100 mM NaCl (blue), 200 mM NaCl (green), and 300 mM NaCl (violet); LAC variety (circles); LAS (squares).

**Table 1 plants-09-00637-t001:** Two-way ANOVA (F values and statistical significance) considering the effect of treatment, variety, and interactions on the measured variables, as indicated. ns, *, **, *** indicate non-significant or significant at *p* < 0.05, *p* < 0.01, and *p* < 0.001, respectively.

**Parameter**	**Treatment**	**Variety**	**Treatment * Variety**
Electric conductivity (EC)	543.28 ***	0.88	0.58
Substrate moisture (SM)	843.73 ***	1.98	3.23
Root length (RL)	9.88 ***	4.23 *	2.89 *
Stem elongation (SE)	12.78 ***	0.89	0.84
Stem diameter increment (SD)	49.55 ***	3.39	0.27
Increment in branch number (Bno)	26.4 ***	0.26	1.10
Root fresh weight (RFW)	20.01 ***	0.14	1.10
Root water content (RWC)	122.12 ***	0.11	0.30
Stem fresh weight (SFW)	44.02 ***	1.01	0.85
Stem water content (SWC)	60.44 ***	2.08	2.87 *
Leaf fresh weight (LFW)	68.59 ***	0.39	0.80
Leaf water content (LWC)	23.06 ***	0.09	1.50
Clorophyll a (Chl a)	9.62 ***	1.27	0.66
Clorophyll b (Chl b)	7.56 **	3.40	1.72
Total carotenoids (Caro)	4.56 **	0.54	1.01
Sodium in roots (Nar)	27.77 ***	0	0.97
Sodium in stems (Nas)	29.67 ***	0	0.10
Sodium in leaves (Nal)	76.44 ***	2.54	2.85 *
Chloride in roots (Clr)	202.93 ***	3.54	5.40 *
Chloride in stems (Cls)	54.16 ***	0.89	0.97
Chloride in leaves (Cll)	140.71 ***	1.53	1.63
Potassium in roots (Kr)	18.59 ***	0.01	0.75
Potassium in stems (Ks)	9.86 ***	0.19	0.84
Potassium in leaves (Kl)	2.24	0.12	0.09
Calcium in roots (Car)	24.09 ***	14.19	8.49
Calcium in stems (Cas)	22.69 ***	2.31	0.64
Calcium in leaves (Cal)	20.20 ***	2.74	0.67
Proline (Pro)	17.07 ***	6.36	2.89
Total soluble sugars (TSSs)	0.73	0.75	1.69

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
