# Peer review of "Effects of Drought and Salinity on Two Commercial Varieties of Lavandula angustifolia Mill"

_plants, 2020, doi:10.3390/plants9050637_

Round 1
Reviewer 1 Report
Dear Authors,
L37 - In many areas of the world?
L47 - ... defense proteins (e.g., - You have forgotten to close the bracket. Maybe e.g is not required?
L79 - LAC and LAS are not explained before. Please explain it before using it. Maybe you can add to L69?
What could be the reason that substrate moisture is high than control.
Figure 1 - different lowercase letters: Then it would be better to address as different uppercase letters instead of capital letters. Please correct all figure legends.
L126 -FW is not defined as fresh weight
L141 - Is there any statistical difference between 2 varieties in 200mM NaCl? If so Capital letters indication is correct? If there is no statistical difference, How you support to your suggestion in L141?
L162-Why Chl b is increased in 300mM NaCl? could not find reason in the discussion.
L235-What could be the reason for low pro content in LAS
PC1 PC2 axis should be indicated in the graph for easiness.
L434 -Please include a reference for WET-2 sensor where a similar application was performed with it.
Author Response
Dear Reviewer,
Thank you for your comments concerning our manuscript entitled “Effects of drought and salinity on two commercial varieties of Lavandula angustifolia Mill.”. Those comments are all valuable and very helpful for revising and improving our manuscript. We have studied your comments carefully and we have made the corrections which we hope will meet with approval.
Best regards,
Zsolt Szekely-Varga

Reviewer 2 Report
I liked the way the paper is written; clearly, well designed experiments, justified conclusions, readable graphs and broad discussion. I only suggest to check minor errors as follows: L47 – remove „(e.g.,” L276 – “Eigenvalue” start writing with lowercase letter L366 why water deficit is in italic? L 381 – “decrease” – use normal font L425 – improve “23oC” L522- probably add https://doi:org/ and put in blue “10.5897/AJB2015.15017.” L530- probably add https://doi:org/ and put in blue “10.1007/s10535-016-0700-9” L538 – there is no Doi: L542 – put in bold “2019” there is no Doi: L550- probably add https://doi:org/ and put in blue “10.1007/s12298-017-0462-7” L554-559 – there is no Doi: L556 and L559 - “2016” should be in bold L567- “plx009” what it is? L581 – in bold “2020” L588- probably add https://doi:org/ and put in blue “10.17660/ActaHortic.2013.988.6.” L 590 – in bold “2012” there is no Doi: L600 –probably add https://doi:org/ and put in blue “10.1007/s11046-005-0206-z” L601- there is no Doi: L606 - probably add https://doi:org/ and put in blue “10.26327/RBL2018.192” L609-617 – there is no Doi: L611 – put in bold “2003” L613 – in bold “2017” L638- probably add https://doi:org/ and put in blue “10.1007/s00248-007-9237-y” L648-651 – add Doi: L662- add Doi: L699 – in bold “2012” L714 – probably add https://doi:org/ and put in blue “10.15389/agrobiology.2018.3.539eng.” L718-725 – there is no Doi: why? L729 – wrap text
Author Response

(The authors gave the same response as above.)

Reviewer 3 Report
The article is written concisely and clearly. The Authors undertook research on the problem of drought, which is very important. Each section is well prepared and contains all necessary information for the reader.
In figure 2 a and b suggests the unification of the unit of length. This will give a better view of plant organ growth in length. In figure 4 for chlorophyll a and b. In figure 5 a common unit for Na (2000), for Cl (900). The same for Ca in figure 6. I also propose to unify the figure in terms of border. Removing the border from figures 1, 7.
For Table 1, I suggest removing horizontal lines from the table and entering full names in the first column with abbreviations in brackets. This will not only make it transparent, but will shorten the title of the table, which in this form disappears among the explanations of the abbreviations, and on the other hand the full names in the table will make it easier to read the results of the parameters.All data in the table should be unified to two decimal places.
I suggest checking of paper according to the author guidelines. Some mistakes I marked in the text (especially in reference part).
After minor revision, I recommend this article for publication.

Author Response

(The authors gave the same response as above.)

Reviewer 4 Report
My comments are very minor.
The manuscript is very well put together.
I have attached a file with some additional comments/suggestions.
I do feel another read over for English and Grammar would be useful. Corrects here should be very minor.
There is some discussion in the results section and this should be removed or the results and discussion section combined.
FW are also discussed at length in the results section, DW or dry mass would seem to be more important and would suggest removal of sections just talking about FW?

Author Response

(The authors gave the same response as above.)
